# GERD—A Burning Problem after Sleeve Gastrectomy?

**DOI:** 10.3390/ijerph182010829

**Published:** 2021-10-15

**Authors:** Przemysław Znamirowski, Piotr Bryk, Piotr Lewitowicz, Dorota Kozieł, Stanisław Głuszek

**Affiliations:** 1Medical College, Jan Kochanowski University in Kielce, 25-369 Kielce, Poland; bryk.piotr@gmail.com (P.B.); piotr.lewitowicz@ujk.edu.pl (P.L.); dkoziel@ujk.edu.pl (D.K.); sgluszek@wp.pl (S.G.); 2Clinic of General, Oncological and Endocrinological Surgery, Provincial Hospital in Kielce, 25-736 Kielce, Poland

**Keywords:** LSG, GERD, EGD, complications

## Abstract

Background: Surgical treatment is the most effective method of treatment for obesity; and laparoscopic sleeve gastrectomy (LSG) is the most frequently performed bariatric surgery. Objective: The aim of the study was evaluation of the frequency of occurrence and the degree of progression of changes characteristic of GERD in patients who had undergone LSG in clinical; endoscopic; and microscopic images in the obtained bioptats; and an attempt to correlate the results obtained with the effectiveness of bariatric treatment. Materials and Method: The anonymized clinical data concerning 214 patients who had undergone LSG were collected from the database. Invitations for check-up examinations were distributed, to which 37 patients responded. Two patients were excluded from the study. In 35 patients after LSG check-up clinical examination, bariatric endoscopy (BE) and bariatric microscopy (BM) were performed on bioptats collected from the site of the gastro-esophageal junction; and 3 specimens collected at 2 cm intervals from the Z-line. The study was performed according to the standard protocol. Results: In the analyzed group, clinical symptoms of GERD occurred in 12 patients (34.5%), including 10 females and 2 males. The symptom reported by all patients was retrosternal pain/burning (heartburn). In BE, foci of ectopic mucosa in the epicardial part of the esophagus were found in 16 patients (14 F and 2 M). No correlation was observed between the analyzed parameters and the occurrence of the above-mentioned changes. In BM, only in three patients were the changes described as normal esophageal mucosa; while in another three, as foci of intestinal metaplasia, Barrett’s esophagus. In this group no foci of dysplasia were found. In eight patients, the changes were described as inflammatory. In ten patients from this group, microscopic changes occurred without clinical symptoms of the disease. Conclusions: GERD is an important clinical problem in patients after LSG; therefore; the problem of occurrence or exacerbation of symptoms of the disease should be discussed with the patient during qualification for bariatric surgery. The bariatric effectiveness of LSG does not correlate with the occurrence of the symptoms of GERD after the procedure. However; the lack of clinical symptoms of the disease does not mean the lack of its occurrence. Therefore; the endoscopic check-up after LSG should be routinely performed. During the qualification for LSG screening, histopathologic examinations of the esophagus may be useful for the assessment of the microscopic symptoms of GERD in oligosymptomatic patients; and exclusion of rare pathologies of the esophagus (e.g., eosinophilic esophagitis), which may complicate post-operative course.

## 1. Introduction

Obesity is a disease of developed countries, and its complications account for an increasing share of direct and indirect costs of the functioning of health care systems. This is a chronic disease associated with an increased morbidity and mortality due to malignant cancer, cardiovascular diseases, diabetes, arterial hypertension, cerebral stroke, and locomotor system diseases. Thus, it leads to disability, and significantly reduces human life expectancy [1]. An excess of energy resulting from the disturbed balance between supply and demand is associated with complex interactions between genetic factors, environmental, and genetic factors, and an individual pattern of behavior of an obese person. It is estimated at more than one tenth of the word population are obese, including also children and adolescents [2]. Obesity is considered as the fifth leading cause of death worldwide. Annually, nearly 3.4 million people die due to this disease [3].

The treatment of obesity is a difficult issue that requires a multidisciplinary approach to the problem. The basic methods applied in the treatment of obesity include: dietary procedure, pharmacological treatment, surgical treatment, rehabilitation and physiotherapy, as well as interventions and changes in the pattern of behavior (behavioral therapy) [4].

The most commonly applied indicator describing the degree of obesity is the Body Mass Index (Quetelet’s index, Body Mass Index, BMI). This index is the ratio between body weight and height expressed in units of kg/m^2^. According to the World Health Organization (WHO), obesity is diagnosed at the BMI value of over 30 kg/m^2^. Table 1 presents the WHO classification of obesity based on the Quetelet’s index.

According to the Polish guidelines, patients with BMI > 40 kg/m^2^ or >35 kg/m^2^ with a concomitant complication of obesity are qualified for surgical treatment [5]. In this group, surgery is the most effective method for treatment of obesity providing, apart from the effect of body weight loss, also an improvement of biochemical and metabolic parameters, such as glycemic profile, lipidogram parameters, or leptin activity [6].

In 2016, in Poland, bariatric surgery was performed in 27 surgical wards. More than 99% of the procedures are performed by laparoscopy. Table 2 presents the percentage of the procedures performed [7].

The procedure most often performed in the Clinic of General, Oncological and Endocrinological Surgery at the Provincial Hospital in Kielce is also laparoscopic sleeve gastrectomy.

Historically, the procedure of sleeve gastrectomy most frequently performed in Poland was the first stage of the procedure of gastric reduction duodenal switch performed in two stages. Accidentally, because some patients disqualified for the second stage due to other causes (apart from surgical), a high effectiveness of the procedure applied was observed. Such a management, much simpler than other surgeries and burdened with a lower risk of long-term complications has become an attractive alternative and, over time, the basic method of treatment for obesity. The procedure consists in freeing the stomach from the greater omentum on the level of 4–6 cm from the pylorus to the angle of His. Subsequently, after insertion through the mouth of the calibration probe of the diameter of 32–40 Fr, a sequential stapling is performed cutting off the greater curvature of the stomach. Electively, some surgeons sew along the stapler seam line. The following benefits resulting from the performance of LSG are mentioned:maintaining the continuity of the duodenum and the small intestine;more stable pharmacokinetics of drugs, especially in patients after transplantation and with circulatory failure;possibility to constantly take medications of the group of non-steroid anti-inflammatory drugs in patients who require their supply, which is a good therapeutic option for patients with connective tissue diseases;however, there may be gastric acid, vitamin B12, iron, magnesium, and ascorbic acid deficiencies, and performance of LSG in most cases does not expose patients to disturbances in absorption of vitamins and trace elements;comparing with gastric binding, no technical problem occurs after LSG procedure, similar to band migration or erosion of the band into the stomach tissue;LSG is in no way inferior from the aspect of results assessed for the reduction of body weight, insulin resistance, or arterial blood pressure to other, more complex methods of surgical treatment of obesity.

Among disadvantages of this procedure mentioned are: irreversibility, a long surgical suture conducive to the occurrence of difficult-to-treat leaks, and postoperative strictures.

GERD (Gastro Esophageal Reflux Disease) is understood as retrograde flux of the gastric contents into the esophagus. It is no coincidence that GERD and obesity are related. In developed countries, the incidence of GERD increases with the frequency of occurrence of obesity. This relationship depends on the frequency of occurrence of GERD complications, especially Barrett’s esophagus.

Visceral adipose tissue secretes leptin, which favors the intensification of reflux of gastric contents into the esophagus. Obesity intensifies esophageal motility disorders, and in manometric tests an increased number of transient lower esophageal sphincter relaxations are observed. Abdominal obesity increases pressure in the abdomen and intensifies the pressure gradient across the diaphragm, which, in turn, is conducive to the formation of hiatal hernias of the diaphragm [8].

It is commonly accepted that gastric banding and sleeve resection do not improve, and even intensify the symptoms of GERD in operated obese patients. Gastric bypass surgery of Roux-en-Y type, apart from the excellent bariatric outcome, has also a good effect in the form of reduction of the intensity of the symptoms of GERD [9]; therefore, such a surgical management is recommended as the treatment of choice in obese patients with GERD.

Patients who had undergone LSG are characterized by reduced lower esophageal sphincter [10]. In association with the performance of resection of the fundus of the stomach, occurs loss of valve anatomical function within the His angle. Tightened sleeve reduces the pressure gradient between the esophagus and the stomach. Also, a tendency is observed towards migration of the left stomach into the chest area. The loss of plasticity of the stomach after the procedure leads to induction of more frequent relaxations of the lower esophageal sphincter, and this mechanism is especially intensified during the first year after surgery. In imaging tests, a tendency is observed towards relative impairment of emptying the proximal part of the stomach and increasing the emptying of the antrum. Too large and stretched sleeve results in an increased secretion of the gastric acid promoting acid reflux. In turn, an excessively narrow, tight sleeve disturbs the outflow of gastric contents also promoting its reflux to the esophagus. This is confirmed by clinical observations, where a larger number of patients after LSG show the occurrence of the symptoms of GERD, or their intensification [8,9,10]. The lack of surveillance and appropriate treatment in such patients may result in an increased frequency of the occurrence of the complications of gastro-esophageal reflux disease with dysplasia leading to disorders including gastro-esophageal junction cancer.

## 2. Objective

The aim of the study was evaluation of the frequency of occurrence and the degree of progression of changes characteristic of GERD in patients who had undergone laparoscopic sleeve gastrectomy in clinical endoscopic and histologic images of bioptates obtained during endoscopy, and an attempt to correlate the results obtained with the effectiveness of bariatric treatment.

## 3. Materials and Method

The consent for the study was obtained on 5 July 2019 from the Bioethical Committee, Collegium Medicum, Jan Kochanowski University in Kielce (No. of consent 43/2019).

From the database of the Clinic of General, Oncological and Endocrinological Surgery at the Provincial Hospital in Kielce, data was downloaded concerning 214 patients who had undergone bariatric surgery during the period from November 2007–March 2019, and they were invited by phone to participate in an endoscopic examination. The patients who reported voluntarily were informed about the aim of the study, and signed a written consent for examinations. Two patients were excluded from the study due to the lack of possibility to perform the planned procedure (lack of cooperation while performing the examination). In one patient, the cause of failure was an excessive excitability and gag reflex during the examination, which made it impossible to collect specimens, whereas in the second case, it was the patient’s active alcoholism and the features of withdrawal syndrome during the examination. The examination was performed according to the standard protocol, without the use of special staining and imaging techniques. Macroscopic assessment was performed using the same Pentax HD endoscopy column and Pentax gastrofiberoscopes.

The following characteristics collected during medical history taking were statistically analyzed: age, gender, maximum body weight before surgery, minimum body weight after surgery, time which had elapsed from the performance of the procedure, body weight while performing examination, and concomitant diseases with particular consideration of the metabolic syndrome. Statistical analysis of the characteristics was performed using arithmetic mean, median range of values (min.-max.), and statistical relevant tests by means of the software STATISTICA version 13.3 Statsoft.

During endoscopic examination, the anatomy of the stomach was assessed, with particular consideration of the fragment of the stomach left in the form of the diverticulum or dilatation of the cardiac part of the stomach, and surgical suture line assessment. Endoscopic changes were assessed within the gastric mucosa. Urease test was routinely performed during the examination. Macroscopic changes indicating GERD were assessed, such as erosions and ulcers (based on grading according to the Los Angeles classification—Table 3) within the Z-line, irregularity within the line with tongue-like protrusions above 2 cm, and foci of macroscopically changed gastric mucosa. The area of the Z-line was also assessed under polarized blue light.

In each patient undergoing the examination, irrespective of the changes observed macroscopically within the mucosa, specimens were routinely collected (from the antrum and the body) for the CLO test, and 4 specimens from the level of Z-line, and proximally at the distance of 2, 4, and 6 cm from the Z-line. In addition, specimens were collected from changes requiring verification for other indications.

The specimens were assessed at the Department of Clinical and Experimental Pathomorphology Collegium Medicum, at the Jan Kochanowski University in Kielce. All tissue specimens were fixed for 12–24 h in buffered 4% formaldehyde. After dehydration, the samples were embedded in paraffin, and 4 micrometer sections were prepared for microscopic evaluation. Routine staining was performed using classic hematoxylin and eosin staining. Microscopic assessments were performed by experienced pathologists. Immunohistochemical methods were applied for detailed assessment. After incubation with primary antibodies (incubation time and temperature in accordance with the producer’s recommendations), further routine activities were carried out. The Ventana ultra View Universal DAB Detection Kit (Ventana Medical Systems; Roche Group, Tucson, AZ, USA) was used as the second antibody. All stages were carried out on the Benchmark Ultra platform (Ventana Medical Systems; Roche Group, Tucson, AZ, USA). In order to diagnose intestinal metaplasia, except for goblet cells metaplasia, expression was sought of CK20 filament proteins typical of the intestinal mucosa and CDX2 gene expression. Goblet cells were detected by means of staining with Alcian blue.

## 4. Results

Thirty-seven patients after LSG reported for examination; two patients were disqualified for the above-mentioned reasons. In one patient, due to the complicated course of GERD before the surgery (histopathologic changes from the polyp of the esophagus endoscopically removed), apart from LSG, fundoplication surgery was additionally performed. Analysis included 35 patients, which constituted 28.4% of those who had undergone surgery in the clinic. In the study group, there were 30 females and 5 males. All patients were treated with proton pomp inhibitors as routine treatment after surgery. Table 4 demonstrates the results obtained.

The last two rows in the above-table attract attention in consideration of the effectiveness of the procedure performed, which describe the effect of regaining body weight, the so-called yo-yo effect.

Figure 1 demonstrates the relationship between the rebound effect and time elapsed since surgical procedure.

While evaluating the statistical data in the whole population examined using the Spearman’s rank-order correlation test for the minimum and current body weight, the correlation coefficient obtained for the whole population was 0.6628, and was very similar in the group of females in the study. Interestingly, in the group of males, this relationship was not confirmed; nevertheless, considering the small size of the group, this fact had no significant effect on the assessment of the whole population examined. Based on the equation of a straight line, it may be estimated that in accordance with such a model, after 12 months the body gain would be: −1.3907 + 0.2257 ∗ 12 = 1.32 kg.

In the study group, ten patients were addicted to nicotine (seven females and three males). The effect of nicotinism on the postoperative course was analyzed in the examined group of patients. Statistical data were analyzed using the Mann–Whitney U test (applying continuity correction). The obtained value *p* = 0.597522 did not provide a basis for rejection of the zero hypothesis that the frequency of occurrence of erosive changes is similar in both groups, and consequently, nicotinism is not the factor which exerts any significant effect on the occurrence of GERD in the examined population.

In the study group, in ten patients no concomitant diseases were diagnosed. Table 5 presents the occurrence of diseases in the analyzed group.

A group of patients attracts attention in whom the regression of the symptoms of diabetes and arterial hypertension was noted as a result of bariatric surgery. In this group, a significantly better bariatric outcome was observed, and the loss of body weight expressed by the change of BMI was from 3.42–17.08; median 13.56 kg/m^2^. Also, in these patients no effect of weight rebound was observed (change from 0.32–0.37 kg/m^2^), and this outcome was fixed in time (analysis from 15–27 months after surgery).

The available results of endoscopy performed before the surgical procedure were also analyzed. Table 6 presents the results of EGD prior to surgical treatment.

While evaluating the above-presented data, it is noteworthy that the symptoms of GERD occurring in patients before LSG, despite the introduction of pharmacological treatment, maintained themselves also after the performance of bariatric surgery.

Symptoms of GERD were evaluated from the following three aspects:occurrence of clinical symptoms of the diseaseoccurrence of changes in endoscopic imageoccurrence of changes in histopathologic examination

The clinical symptoms of the disease were classified as: retrosternal burning (heartburn, pyrosis), regurgitation, rumination, dysphagia, salivation, sensation of a lump in the throat, and odynophagia.

In the analyzed group, the clinical symptoms of GERD occurred in 12 patients (34.5% of the examined patients), including 10 females and 2 males. No correlation was found between any of the above-analyzed parameters and their occurrence. The symptom reported by all patients was retrosternal pain/burning (heartburn).

While analyzing endoscopic images, the foci of altered mucosa were observed, as well as transition of the Z-line (so-called tongues), esophageal erosions classified according to the Los Angeles scale, and ectopic foci within the esophageal mucosa.

Table 7 demonstrates the occurrence of esophageal ulcers according to the Los Angeles classification.

Data concerning the occurrence of erosive changes during the period before and after surgery were analyzed using the sign test and the Wilcoxon signed rank test. The *p* value for signs was 0.2278, which authorizes us to state that statistically the distribution of the occurrence of erosive changes in the esophagus was the same during pre-operative and post-operative periods. In the signed-rank test, the p coefficient was considerably lower—0.1096. Assuming the risk of error on the level of 15% (with respect to the standard 5%), assuming significance on the level of *p* = 0.15, it may be seen that the occurrence of erosive changes was more frequent after the surgery (assuming the risk of error on the level of 15%).

Foci of mucosal ectopy in the esophagus supra-channel area were observed in 16 patients (14 females and 2 males). No correlation was found between the analyzed indicators and the occurrence of the above-mentioned changes. In histopathologic analysis only in three patients, the above-mentioned changes were described as normal esophageal mucosa. In three patients, they were described as foci of intestinal metaplasia—Barrett’s esophagus. No foci of dysplasia were found in this group. In eight patients, they were described as inflammatory changes. In ten patients from this group, the changes were asymptomatic.

While analyzing the occurrence of tongue-like protrusions of the Z-line and its irregularities in the examined group, it should be emphasized that in endoscopic evaluation, their size up to 2 cm should be considered as normal. Nevertheless, it was observed that the changes of this type, which fall within the range of the standard may also be accompanied by histopathologic changes. It is also worth noting that in one patient, this was the only endoscopic symptom of the presence of intestinal metaplasia in histopathologic examination. Table 8 presents the results. No statistical correlation was observed between the analyzed indicators and the occurrence of the above-mentioned changes.

The presence of endoscopic symptoms of hiatal hernia was observed in four patients in the form of lowering the impressions of the diaphragm branches, and in four patients—in the form of incompetence of the gastric cardia, with these both features together observed in two patients. In all these patients, other endoscopic symptoms of GERD were observed. Interestingly, in EGD performed before the surgery (if available for analysis) the described symptoms were not found in the examined group of patients. This suggests that the narrowed gastric remnant may migrate to the chest during the post-operative period.

Table 9 demonstrates the results of histopathologic assessment according to the site of specimen collection.

In the detected foci of intestinal metaplasia, no features of the occurrence of dysplasia were found. Nevertheless, it is worth mentioning that this group were patients with at least one endoscopic symptom of GERD. It is also an interesting fact that only one of these patients reported the presence of clinical symptoms of GERD.

In one female patient, eosinophilic esophagitis was diagnosed, which was not observed in endoscopic examination before the surgery. This patient reported clinical symptoms of GERD, and she was also diagnosed with a small linear erosion (LA A). Previously, she was not treated due to concomitant diseases. The course of surgery was uncomplicated; however, in this patient a good bariatric outcome was not obtained. The example of this patient evokes the question concerning the necessity for histopathologic assessment of esophageal mucosa even when unchanged in the endoscopic image during the qualification for LSG.

An example of a female patient who was operated on in an extended manner also deserves attention, i.e., she had undergone LSG procedure extended by fundoplication. Such a qualification was underlain by the complicated course of GERD, and the lack of consent for performance of Roux-en-Y gastric bypass. In this patient, a very good bariatric outcome was achieved within a 16-month observation period, with remission of the symptoms of the metabolic syndrome and decrease in BMI from 46.65 to 33.43, with no yo-yo effect observed. Anti-reflux procedure performed as a revision surgery after bariatrics does not seem to be a sufficient and effective management in the case of GERD taking into account a considerable increase of pressure in the LSG gastric remnant.

Summing up, the voluntary character of patients reporting to the examination may raise the results of post-operative assessment of changes in the esophagus due to stimulation by the occurrence of the symptoms of GERD. In turn, for obvious reasons, referring the results to the whole population brings about the risk of underestimation of the problem. However, it is worth mentioning that the preliminary evaluations of the occurrence of GERD in patients who had undergone LSG are clinically interesting and require further studies and analyses. Table 10 demonstrates the compilation of the frequency of occurrence of clinical, endoscopic, and histopathologic symptoms in the examined group, and with respect to the total number of LSG procedures performed in the clinic.

Analysis of the above-presented data reveals that more than 82% of patients presented at least one symptom of GERD. In 17 patients with endoscopic and/or histopathologic symptoms, clinical symptoms were not observed.

## 5. Discussion

An increase in the frequency of performance of LSG is accompanied by an increasing number of reports concerning the consequences of this surgery. However, while analyzing the literature, it is hard to resist the impression that the reported data are contradictory. The number of possible mechanisms promoting and controlling GERD after the surgery is large, and their contribution, in fact, is unknown. The problem requires extensive, multi-center studies in order to develop a scheme of management of patients after LSG. Similar to the presented study, Praveen Raj et al. [11] concluded that the frequency of occurrence of GERD after LSG is big enough that according to them, GERD should be a contraindication to LSG. Similar to our evaluations, the frequency of occurrence of GERD after LSG in the examined group was 66.6%. During the pre-operative and post-operative periods, the researchers performed 24-h pH monitoring. They observed a significantly larger number of episodes of reflux of acidic stomach contents into the esophagus in the group of patients who had undergone LSG, compared to RYGB. This may explain the observed lack of clinical symptoms typical of acidic reflux, with the occurrence of endoscopic and histopathologic symptoms. This is also confirmed by our observation that some of the patients with asymptomatic GERD may be exposed to equally dangerous alkaline refluxes. Lihu Gu et al. [12] arrived at similar conclusions. The performance of RYGB is more effective in the treatment of the symptoms of GERD, compared to LSG, whereas the frequency of occurrence of new symptoms of GERD is significantly higher in obese patients who had undergone LSG.

A team of French researchers carried out a five-year endoscopic observation of patients after LSG. In their study, Dimbezel et al. [13] confirmed that there is a large number of asymptomatic patients with many findings typical of GERD in endoscopic examination. The researchers indicated that among these patients there are also patients with severe course of GERD, and postulated the conversion from LSG to RYGB in this group of patients, which undoubtedly justifies the necessity for endoscopic surveillance of patients who had undergone LSG. A group of Swiss researchers published results concerning 222 patients after LSG [14]. In the case of intraoperatively found presence of hiatal hernia, the patients were qualified for posterior hernioplasty. In this group, 116 patients presented the symptoms of GERD after surgery, of which, as many as 85 were ‘de novo’. Similar to our observations, the researchers concluded that even more than half of patients after LSG will have the symptoms of GERD, while the majority of preoperatively asymptomatic patients will develop full symptoms of the disease. A team of researchers from Italy [15] was more cautious in their evaluation of the problem. Based on data from 24-h pH monitoring, a reduction in the intensity of the symptoms of GERD was observed in 28 patients who had undergone LSG, whereas the symptoms of GERD ‘de novo’ were noted in only 5.7% of cases, which does not confirm our observations. Interestingly, the researchers postulated that the shape of the sleeve formed from the stomach is of primary importance. They considered it beneficial leaving the dilatation in the proximal part of the stomach (neofundus), leaving a fragment of the antrum, which is supposed to improve gastric emptying, and consequently decreases the symptoms of GERD, using possibly the largest-diameter catheters in order to avoid an excessive narrowing of the stomach lumen during LSG. Also, according to the researchers, the preoperative presence of hiatal hernia and one-step repair requires further assessment in standardized multi-center studies. Hendricks et al. [16] in a group of 919 patients after LSG, observed GERD in only 38 patients (4%), out of whom the symptoms of the disease were ‘de novo’ in 25 patients, and four patients from both groups required conversion to RYGB due to GERD, while the reminder responded to pharmacological treatment with proton pump inhibitors. In turn, Csendes et al. [17] paid attention to the fact that constantly open and immobile pylorus also favors the reflux of duodenal contents into the sleeve and further to the esophagus. Based on the pre-operative assessment, the examined patients were divided into two groups (I non-refluxers and II-refluxers). It was found that in Group 1, in as many as 58% of patients the symptoms of GERD de novo were detected, whereas in Group 2, only in 14% of patients the remission of the symptoms was observed. The researchers stated that LSG should be approached as a procedure conducive to the occurrence of GERD, and patients who had undergone this procedure require constant, regular endoscopic surveillance. This is consistent with the clinically asymptomatic group of patients observed in our study with endoscopic and/or histopathologic symptoms of GERD. While analyzing possible modifications of treatment, Taiwanese researchers described an attempt of treatment of GERD in patients after LSG by means of repair of hiatal hernia with one-step gastropexy. However, despite the lack of substantial surgical complications of the procedure, in their conclusions, they described only a partial remission of the symptoms of GERD [18]. Ece et al. [19] postulated an interesting approach to the problem. They proposed isolation of patients with clinically silent GERD, based on ambulatory pH monitoring, and subsequently the performance of LSG cruroplasty. While analyzing the results concerning a group of more than 400 patients, in 70 of them the features of GERD were observed, while 59 had undergone LSG with cruroplasty; 2 patients in the examined group developed the symptoms of GERD, compared to 11 in the whole group in whom GERD occurred ‘de novo’, despite a good outcome in preoperative pH monitoring.

In a study published in 2020, Chinese researchers summed up the issue of LSG extended by anti-reflux procedures. [20]. They confirmed unequivocally that the effectiveness of the combined methods consisting of combining the interventions of hiatal repair (cruroplasty) or fundoplication with LSG is uncertain, and the variety of the techniques and qualifications applied requires the standardization of data in order to evaluate them. Nevertheless, in their analysis the researchers pointed out that in cases of patients with GERD complicated after LSG, RYGB is a more effective therapeutic option than additional anti-reflux procedures, or repeated extended LSG. Also in our observations, the performance of one-step anti-reflux procedure was not associated with the remission of the symptoms of GERD in a patient after LSG.

GERD is a very serious clinical problem in patients after LSG. While qualifying a patient for bariatric treatment and establishing the type of the procedure performed, the advantages and disadvantages of each type of management should be discussed with the patient, considering the risk of postoperative complications. The possibility of the occurrence of GERD and its complications, and associated with this decrease in the quality of life is undoubtedly an important element of such a conversation. The simplicity of performing LSG and a relatively small number of complications, as well as very good effectiveness in the metabolic context incline surgeons and patients to undertake this therapeutic option. It should always be kept in mind that a patient requires careful qualification for the surgery, and equally careful surveillance after the procedure, for possible early detection of complications and their proper treatment. The surveillance of the patient and the need for this surveillance, similar to the modification of health behaviors, is inscribed in the whole change of the patient’s pattern of behavior. Physicians providing care to obese patients must also always remember that in the first place they should do no harm.

It is worth noting that scientists are constantly looking for new minimally invasive therapeutic options. One of them is endovascular bariatric left gastric artery (LGA) embolization [21]. Although transient gastric ulceration has been described as one of the complications of surgery, there are no data on its long-term effects, including its effect on GERD. It cannot be ruled out that the lack of mechanical narrowing of the stomach may reduce the occurrence of GERD to a simple correlation with the patient’s body weight.

Summing up: GERD is an important clinical problem in patients who had undergone LSG, and the bariatric effectiveness of LSG does not correlate with the occurrence of the symptoms of GERD after the surgery. The absence of clinical symptoms of the disease does not mean the lack of its occurrence. In view of the above, endoscopic surveillance together with histopathologic assessment of the esophageal mucosa should be routinely performed after LSG, and in the case of diagnosing the symptoms of GERD, a proper therapeutic management should be introduced. The detected changes were of a mild character; however, the foci of metaplasia were observed even in patients without the symptoms of GERD. It should be expected that in non-treated patients, during further observation, there is a possibility to diagnose dysplasia and cancerous process.

The study was conducted in accordance with the presented methodology, an important clinical problem was analyzed—the occurrence of GERD with the possibilities of long-term consequences after bariatric surgery—LSG. The limitation of the study is a small number of patients. The presented results are undoubtedly an interesting introduction to the expansion of studies of this problem in the future.

## 6. Conclusions

Considering the mean age of patients who undergo bariatric surgery (LSG), clinical observation with endoscopic surveillance should be planned over a long period of time in order to reduce the risk of serious complications of GERD, including gastro-esophageal junction cancer.

While qualifying patients for bariatric procedure extending the routine endoscopic assessment should be considered by the collection of specimens from the epicardial part of the esophagus (especially in groups at risk), even with the lack of the macroscopic image of pathological changes.

Patients qualified for LSG who present the symptoms of GERD should be informed before the surgery about the possibility of intensification of the symptoms of the disease after surgery, and the possible necessity for application of other, even surgical methods, of treatment within a longer time of observation.

## Figures and Tables

**Figure 1 ijerph-18-10829-f001:**
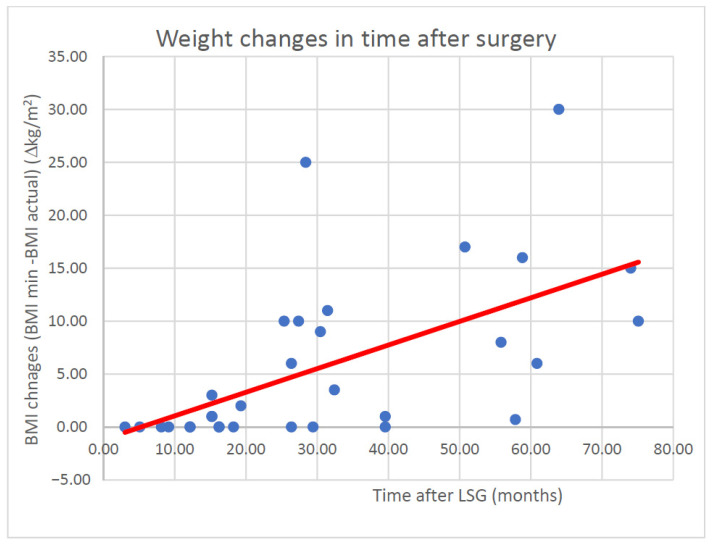
Relationship between change of BMI (yo-yo effect) and time (in months) elapsed since LSG.

**Table 1 ijerph-18-10829-t001:** WHO classification of obesity based on BMI.

BMI (kg/m^2^)	WHO Classification
Below 18.5	Underweight
18.5–24.9	Normal
25.0–29.9	Overweight
30.0–34.9	Grade 1 obesity
35.0–39.9	Grade 2 obesity
>40.0	Grade 3 obesity (also known as morbid, extreme, or severe obesity)

**Table 2 ijerph-18-10829-t002:** Bariatric procedures performed in Poland in 2016.

Name of Procedure	Number of Surgeries Performed	Percentage
Sleeve gastrectomy (LSG)	1032	64.6%
Laparoscopic Roux-en-Y gastric bypass	291	18.2%
One anastomosis-gastric bypass	132	8.3%
Laparoscopic adjustable gastric banding (LAGB)	117	7.3%

**Table 3 ijerph-18-10829-t003:** Los Angeles classification.

Classification	Feature
Los Angeles A	One mucosal break ≤5 mm
Los Angeles B	≥1 mucosal breaks of the length >5 mm, without continuity between 2 adjacent mucosal folds
Los Angeles C	≥1 mucosal breaks continuous between ≥2 mucosal folds, but involving ≤75% of the esophageal circumference

**Table 4 ijerph-18-10829-t004:** Statistical analysis of the examined population.

	Arithmetic Mean	Median	Standard Deviation	Minimum	Maximum
Age (years)	43.14	44.01	8.67	26.49	64.39
Height (cm)	169.95	170.00	8.00	150.00	183.00
Body weight (max) (kg)	130.78	130.00	21.23	80.00	177.00
BMI (max) (kg/m^2^)	45.26	45.53	5.48	35.56	58.82
Body weight (min)	92.02	92.00	17.0	56.00	130.00
BMI (min)	31.92	32.43	5.31	21.83	44.98
Change of body weight (max) (kg)	38.76	36.00	13.89	10.00	65.20
Change of BMI (min) (kg/m^2^)	13.33	13.09	4.24	3.42	21.08
Time elapsed after surgery (months)	30.56	26.40	20.18	3.03	75.10
Change of body weight (min-act) kg	5.51	1.50	7.66	0.00	30.00
Change of BMI (min-act) kg	2.00	0.53	2.92	0.00	11.87

**Table 5 ijerph-18-10829-t005:** Diseases accompanying obesity in the examined population.

Disease	Females	Males	Total
Arterial hypertension	9	3	12
Including the resolution of symptoms after surgical treatment	5		5
Type 2 diabetes	4	1	5
Including the resolution of symptoms after surgical treatment	3		3
Hypothyroidism	4		4
Simple non-toxic goitre	1		1
Osteoporosis	1		1
Depressive disorders	2		2
Cardiomyopathy		1	1
Epilepsy	2		2
Cancerous diseases	1 (breast)		1
Prolactinoma	1		1
Ischaemic heart disease		1	1

**Table 6 ijerph-18-10829-t006:** Results of pre-operative endoscopic assessment in the examined group of patients.

	Females	Males	Total
Esophageal polyp	1	0	1
Esophageal erosions	2	1	3
Hiatal hernia (endoscopic features)	2	2	4
Erythematous gastritis	7	1	8
Positive CLO TEST	8	1	9
Normal endoscopy	5	1	6

**Table 7 ijerph-18-10829-t007:** Erosive esophagitis (*EE-Esophagitis Erosiva*) in post-operative assessment (in brackets the occurrence of EE in pre-operative assessment).

Classification	Females	Males	Total	%
Los Angeles A	7 (1)	1 (1)	8 (3)	22.85
Los Angeles B	1 (1)	1 (0)	2 (1)	5.71
Los Angeles C	0 (0)	1 (0)	1 (0)	2.86
TOTAL	8 (2)	3 (1)	11 (3)	31.42
%	26.66	60	31.42	

**Table 8 ijerph-18-10829-t008:** Irregularities of the Z-line in post-operative endoscopic assessment.

Classification	Females	Males	Total
Protrusion of Z-line ≤ 2 cm	6	2	8
Protrusion of Z-line > 2 cm	2	0	1
TOTAL	8	2	10
%	26.66	40	31.42

**Table 9 ijerph-18-10829-t009:** Results of histopathologic assessment of the collected bioptats of the esophageal mucosa.

Type of Change	0 cm	2 cm	4 cm	6 cm
Esophagitis	13	5	2	0
Intestinal metaplasia (without dysplasia)	4	0	0	0
Eosinophilic esophagitis	0	1	1	1
TOTAL	17	6	3	1

**Table 10 ijerph-18-10829-t010:** Compilation of the symptoms of GERD in the group of patients who reported to the examination, and with respect to the total number of patients who had undergone surgery in the Clinic of General, Oncological and Endocrinological Surgery at the Provincial Hospital in Kielce.

	Females	Males	Total	% of Patients Participating in the Study
Clinical symptoms	1	0	1	2.85
Endoscopic alterations	3	2	5	14.28
Histopathologic alterations	1	0	1	2.85
Clinical + endoscopic symptoms	1	1	2	5.71
Endoscopic + histopathologic symptoms	10	1	11	31.42
Clinical + histopathologic symptoms	0	0	0	0
Clinical + endoscopic +histopathologic symptoms	8	1	9	25.71
Total	24	5	29	82.85
% of study participants	80	100	82.85	x

## Data Availability

Presented studies are purely observational and do not require registration.

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
