# Peer review of "GERD—A Burning Problem after Sleeve Gastrectomy?"

_ijerph, 2021, doi:10.3390/ijerph182010829_

Round 1
Reviewer 1 Report
Well written, sound scientific data.
Few minor suggestions:
- Graph 1 axis labels are not written in english
- I should have expected an important reduction in GERD manifestations after laparoscopic sleeve gastrectomy due to the weight loss. Is not weight loss associated to GERD improvement?
- Describe better the profile of patients with GERD after LSG, maybe with a table
- Cite this article among the new possible alternative options for obesity treatment: Sangiorgi GM et. Nutrients. 2021 Jul 25;13(8):2541. doi: 10.3390/nu13082541. PMID: 34444701; PMCID: PMC8401754.
- The text and the style must be revised, it must be more fluent and readable.
Author Response
Dear Reviewer
On behalf of the team, thank you for reading our manuscript, for a thorough analysis and valuable comments.
I hereby wish to comment on the comments and provide an explanation:
Graph 1 axis labels are not written in English
The graph 1 was changed to be editable in MS Excel format and descriptions in English were introduced.
I should have expected an important reduction in GERD manifestations after laparoscopic sleeve gastrectomy due to the weight loss. Is not weight loss associated to GERD improvement?
One of the conclusions of our work is the currently observed lack of correlation between the occurrence of GERD and weight loss. Changes in the time of LSG, such as the anatomical elimination of anti-reflux mechanisms and increased pressure in the stomach resulting from the narrowing of its lumen, most likely counteract the beneficial effect of weight loss. Therefore, just a good weight loss effect does not exempt from observation towards GERD.
Describe better the profile of patients with GERD after LSG, maybe with a table
The patient's profile is described in Table 4. As mentioned in the text, we have not noticed significant correlations between the occurrence of GERD and any element describing the population undergoing LSG surgery.
For this reason, it is difficult to make any conclusions about the population that may affect the occurrence of GERD. Of course, the limitation is the relatively small group of respondents and the fact that the study was carried out on a group of volunteers whose reporting could result from symptoms, which may affect the overdiagnosis of GERD. However, please note that even in relation to the entire population (including those who did not report) GERD is an important clinical problem.
Cite this article among the new possible alternative options for obesity treatment: Sangiorgi GM et. Nutrients. 2021 Jul 25;13(8):2541. doi: 10.3390/nu13082541. PMID: 34444701; PMCID: PMC8401754.
We got acquainted with the interest in the method of endovascular treatment and we mentioned it in the discussion about works on this type of therapy.
The text and the style must be revised, it must be more fluent and readable.
We are doing our best and the paper was translated by professional translator.
Thanks again for your comments. Best regards from Kielce
Reviewer 2 Report
I read with interest the manuscript entitled "GERD - a burning problem after sleeve gastrectomy?". I think this is an interesting paper with a complete background, describing results in detail and a good discussion of literature.
I have only some concerns and comments:
One of the classical symptoms of GERD is regurgitation. Any of the patients had this symptom or it was not considered at all?
In the Introduction it is stated that sleeve gastrectomy does not lead to micronutrients deficiencies. I do not agree with this point, as the lack of gastric acid secretion may lead to iron or cobalamin deficiency and in some cases also to malabsorption of magnesium or ascorbic acid. This should be corrected.
In the final paragraphs of the Introduction section statements are reported without references.
Results: I would suggest subdividing the Results section into some subheadings to increase readability.
The time after surgery, at which endoscopy was performed was variable ranging from 3 to 75 months. This might be discussed as the percentage of GERD may be underestimated in those with very short follow-up.
There was no mention of treatment, for example, antisecretory drugs after surgery which are very often prescribed. This should be discussed as these drugs do not work in patients with sleeve gastrectomy due to impaired gastric acid secretion and pharmacological treatment of GERD might be challenging in these patients.
Table 10: the expression "endoscopy symptoms" and histologpathological symptoms" sounds odd. Did you mean signs? changes? alterations"
Author Response
Dear Reviewer
On behalf of the team, thank you for reading our manuscript, for a thorough analysis and valuable comments.
I hereby wish to comment on the comments and provide an explanation:
One of the classical symptoms of GERD is regurgitation. Any of the patients had this symptom or it was not considered at all?
We considered regurgitation, but none of our patients reported this problem. The only clinically reported symptom of GERD in our group was burning behind the breastbone. This symptom affected 10 people.
In the Introduction it is stated that sleeve gastrectomy does not lead to micronutrients deficiencies. I do not agree with this point, as the lack of gastric acid secretion may lead to iron or cobalamin deficiency and in some cases also to malabsorption of magnesium or ascorbic acid. This should be corrected.
Thank you for your comment. We have introduced information. However, this does not change the fact that compared to procedures such as Roux-en-Y gastric bypass, LSG causes deficiency symptoms very rarely.
In the final paragraphs of the Introduction section statements are reported without references.
We have made changes. Part of the problem was due to technical reasons and text editing after uploading to the server
The time after surgery, at which endoscopy was performed was variable ranging from 3 to 75 months. This might be discussed as the percentage of GERD may be underestimated in those with very short follow-up.
This is obviously one of the limitations of the study, but our analysis shows that GERD also occurred during this short period of observation even in patients who had no symptoms of GERD before the surgery. Additionally, the occurrence of GERD was not dependent on the weight loss effect. This is probably due to the reduction of the beneficial effect by the natural changes that the patient undergoes during LSG, such as reduction of the Hiss angle or high pressure in the constricted lumen of the stomach.
There was no mention of treatment, for example, antisecretory drugs after surgery which are very often prescribed. This should be discussed as these drugs do not work in patients with sleeve gastrectomy due to impaired gastric acid secretion and pharmacological treatment of GERD might be challenging in these patients.
Our patients routinely in the postoperative period are treated with a proton pump inhibitor. Information on this subject has been included in the text. This fact may explain why some patients did not develop clinical symptoms of GERD in the presence of endoscopic and histopathological alterations.
Table 10: the expression "endoscopy symptoms" and histologpathological symptoms" sounds odd. Did you mean signs? changes? alterations"
We made the suggested fixes
Thanks again for your comments. Best regards from Kielce